# Mouse Models for *Mycobacterium tuberculosis* Pathogenesis: Show and Do Not Tell

**DOI:** 10.3390/pathogens12010049

**Published:** 2022-12-28

**Authors:** Pablo Soldevilla, Cristina Vilaplana, Pere-Joan Cardona

**Affiliations:** 1Unitat de Tuberculosi Experimental, Germans Trias i Pujol Research Institute (IGTP), 08916 Badalona, Spain; 2North Metropolitan Clinical Laboratory, Microbiology Department, ‘Germans Trias i Pujol’ University Hospital, 08916 Badalona, Spain; 3Genetics and Microbiology Department, Universitat Autònoma de Barcelona, 08913 Cerdanyola del Vallès, Spain; 4Centro de Investigación Biomédica en Red de Enfermedades Respiratorias (CIBERES), 28029 Madrid, Spain

**Keywords:** tuberculosis, mouse, C3HeB/FeJ, vaccines, immunopathology, chemotherapy, tolerance, resistance, history

## Abstract

Science has been taking profit from animal models since the first translational experiments back in ancient Greece. From there, and across all history, several remarkable findings have been obtained using animal models. One of the most popular models, especially for research in infectious diseases, is the mouse. Regarding research in tuberculosis, the mouse has provided useful information about host and bacterial traits related to susceptibility to the infection. The effect of aging, sexual dimorphisms, the route of infection, genetic differences between mice lineages and unbalanced immunity scenarios upon *Mycobacterium tuberculosis* infection and tuberculosis development has helped, helps and will help biomedical researchers in the design of new tools for diagnosis, treatment and prevention of tuberculosis, despite various discrepancies and the lack of deep study in some areas of these traits.

## 1. Introduction

The use of animals for scientific purposes is something that has been valued in almost all the ages of mankind’s history. The first described experience, by some authors, is dated in the fourth century BC in ancient Greece [1,2,3]. From then on, and until the nineteenth century, animal experimentation was practiced, especially in all of Europe, and it was mainly focused on the same goals as in ancient Greece, i.e., to generate useful anatomical knowledge for human medicine [4,5]. 

During the second half of the nineteenth century, Louis Pasteur and Robert Koch popularized the use of animals to explore the causes of infectious diseases leading to the definitive establishment of the germ theory of infectious diseases in society [6]. Their work and methodology with animals not only provided a great advance in public health for the results and knowledge generated from them, but it also served as a model for conducting animal research related with microbial pathogens, impregnating the minds of novel researcher generations, especially Americans [7].

Asia brought to Europe and the rest of the world the knowledge of breeding rodents [8,9], which allowed their use as new potential laboratory animal in the beginning of the 20th century [10,11]. Mice and rats grew in popularity as experimental models throughout the twentieth century until they finally surpassed other non-rodent animal models, such as dogs, rabbits, swine, etc., in terms of number of scientific publications, with mice being the most used animal model [12].

William Ernest Castle, together with Clarence Little, developed the first inbred mice lineage [13]. Later, with the financial support of Roscoe Jackson and Edsel Ford, Little founded his own laboratory dedicated to mice breeding, which would end in what we know today as “Jackson Laboratory”, developing most of the inbred mice lineages that are used today [14]. 

The methodology and experiences of Pasteur and Koch combined with the potential of rodents allowed a deep study of human infectious diseases using these animal models [15].

From among all infectious diseases, the relationship between tuberculosis and humans is probably the oldest of all [16,17].

It is necessary to remark that the discovery of *Mycobacterium tuberculosis* by Robert Koch as the causative agent of tuberculosis was made using the guinea pig, another rodent, as an animal model [18]. This finding would be the beginning of a new and prosperous era of biomedical research on tuberculosis using animal models, rodents overall, to decipher the pathology of the disease, its way of transmission and how we could respond against it. In this review, we try to show some of the most notable advances achieved in tuberculosis research, including several aspects related with development of the disease, diagnosis, prophylaxis and therapeutic treatments, having used animal models and focusing mainly on the mice model. We will also discuss the advantages and disadvantages of the mice model in tuberculosis research, as well as the aspects and considerations to take into account in experiment design using mice, and the future perspectives of animal experimentation in tuberculosis research.

## 2. Tuberculosis: Knowing the Enemy

### 2.1. M. tuberculosis Infectious Pathway

As Sun Tzu pointed out in *The Art of War*: “If you know the enemy and know yourself, you need not fear the result of a hundred battles” [19], meaning that it is necessary to know and understand all the capabilities and weaknesses of the enemy you are confronting, as well as your capabilities and weaknesses, in order to win the battle against it. If we do a simile, we can consider tuberculosis as “the enemy” to beat, the human host as “yourself” and the actions between the “two armies”, i.e., *M. tuberculosis* as tuberculosis soldiers and the cellular defense mechanisms as the human host soldiers, as “the battle”.

It is mandatory, for the purpose of understanding a disease, to study and describe its cycle in the affected host. In the case of tuberculosis, as it is an airborne-transmitted disease, everything starts when aerosols charged with *M. tuberculosis* from an infected person arrive to the lung of a potential new host. The bacillus is deposited in the alveoli, where it is firstly attacked by pulmonary surfactant, secreted by type II pneumocytes. The properties and composition of the surfactant can cause physical damage to the cell wall of *M. tuberculosis*. However, the bacillus is able to detach lipids from its wall that can act as a countermeasure to the surfactant effect by changing its composition and quality, which allows the bacteria to avoid further damage [20]. Pulmonary surfactant is also able to enhance the interactions between the bacteria and the alveolar macrophages [21]. Alveolar macrophages (AM) are tissue resident cells with great phagocytic capacity, being the first direct contact of cellular defense against *M. tuberculosis* and other bacteria, virus or fungus that arrives to the lung. They also secrete cytokines and chemokines to attract other immune cell populations to the infection focus [22,23]. 

The infection progresses once *M. tuberculosis* is phagocyted by the AM. There, it produces a peptide called ESAT-6, which inhibits the maturation of the phagolysosome and gives the bacteria access to the nutrient-rich cytosol of the macrophage. The bacteria start to replicate and grow in number until they cause necrosis of the macrophage, then gaining access to the extracellular environment [24,25]. During intracellular growth, *M. tuberculosis* starts forming bacterial aggregates that receive the name of cords. The formation of this clump will benefit the progression of the infection in future phases and has been associated with increased virulence in human infection [26].

These “new” bacteria are targeted by other AMs, making the cycle of phagocytosis-inhibition–replication–necrosis repeat again. During this cycle, infected and necrotic macrophages release inflammatory chemokines that attract polymorphonuclear cells (PMNs), especially neutrophils, and monocytes to the infection focus. Monocytes will differentiate in macrophages once they arrive to the lung. This will stimulate the replication of *M. tuberculosis* in the same way that occurs with AMs. Neutrophils also have phagocytic capacities, but this function is only effective when they encounter small size pathogens and/or bacterial aggregates. If the size of the biological threat overcomes the phagocytic capacities of neutrophils, they will develop neutrophil extracellular traps (NETs), which are accumulations of DNA and proteins released by neutrophils via cell death or degranulation. NETs have an antimicrobial effect upon some infections, but they also can damage the host’s tissues if this response is dysregulated [27,28]. That is exactly what happens when neutrophils face *M. tuberculosis* aggregates (cords, previously commented). Incapacity of phagocyting the cords leads to the development of NETs and pulmonary tissue destruction, especially under hypoxic conditions (e.g., areas where the bacteria have already consumed all the available oxygen), but it also seems to help macrophages to deal with the infection [29,30]. It also has been reported that cording formation can stimulate the formation of macrophage extracellular traps (MET), but this phenomenon needs to be studied more [31]. This inflammatory response also enhances the drainage of affected alveoli to the lymph nodes, where dendritic cells process *M. tuberculosis* antigens and initiate the adaptive immune response [32].

Upon activation of CD4 T cells by antigen presentation from dendritic cells, two possible immunological routes can be driven in the host: (1) If an unregulated Th17 response is developed, IL-17 secreting T cells will recruit more neutrophils and other PMNs to the infection, dysregulating the inflammatory response and causing the increase in the lesion and severe tissue destruction, as a result of uncontrolled proliferation of NETs, ending in the infection disarray and bacterial dissemination through the lung or, in some cases, reaching extrapulmonary systems of the host; however, (2) if a coordinated response between Th1 and Th17 cells is developed, IFN-γ producing T cells will activate the infected macrophages to augment their phagocytic capacity. This will lead to the control of bacillary population, and will favor the encapsulation of the lesion, avoiding its progression. Macrophages then generate intracellular lipid droplets from phagocyting the bacilli and cellular debris that give them a “bubble-filled” aspect, so they receive the name foamy macrophages [33,34]. 

Th2 response is also involved in containing the infection, in a coordinated way with Th1 response, by avoiding an excessive inflammatory response [35], but it has been described how an excess of Th2 response could enhance the progression of the infection [36]. Regulatory T-cells also play an important role in controlling the disease, as these cells are in charge of maintaining the necessary balance between Th1 and Th17 responses [37]. The contribution of B cells and antibody production in controlling the infection is not clear yet. This response seems to have evidences for either a positive or a negative role, reviewed elsewhere [38,39].

Development of granuloma, the most characteristic structure of the disease, takes place in both scenarios explained above, in order to contain the infection—or, at least, trying to—but its cellular composition, organization and effectiveness in that task will be different between immunological environments [40,41]. If the immune response fails in containing the infection, the rest of the pulmonary tissue is compromised. Bacterial dissemination will enhance the formation of other lesions in the lung, which eventually could allow the bacteria to arrive to the bloodstream and colonize other organs and structures, leading to the systemic failure and death of the host if medical interventions are not performed [42] (Figure 1).

On the other hand, after the initial infection, there is also the possibility that the infected individual remains infected for years without developing the disease, what is known as latent infection [43]. Nowadays, there are two hypotheses to explain why this happens.

The classical view argues that, in this phase or stage, the bacteria adopt a metabolically inactive state in which it is unable to replicate, so it cannot cause the disease and the immune system does not detect it. It is thought that these inactive bacilli could persist inside granulomas “waiting” for the appropriate conditions, such as new exogenous *M. tuberculosis* infection or immunosuppression of the host, conditions that would relieve the containment pressure of the already existing granuloma and would allow the inactive bacteria to reactivate and cause disease again [44,45].

However, in recent years, another hypothesis has been purposed to substitute the former: the dynamic hypothesis. In this theory, latent infection is described as a constant reinfection process that occurs while the metabolic inactive mycobacteria are drained in alveolar fluid through the respiratory tract until reaching the gastrointestinal duct. It has been suggested that the mycobacteria, during this drainage, can also abandon the initial granuloma inside one of the most characteristic tuberculosis pathology cells, Foamy macrophages, which are AMs that, after phagocyting bacterial cells and cellular debris, develop internal lipid droplets. After leaving the granuloma, the bacteria can find new pulmonary niches to start a new infectious focus [46,47,48,49,50].

Transition or reactivation from latent to active infection and development of active tuberculosis occurs mainly during the first year after infection, and the risk decreases exponentially. Progression depends on several host- and not host-related factors, such as immune status, environmental conditions or mycobacterial characteristics, with an average probability between 5−15%, although this probability is higher in infants with less than 14 years of age [51,52,53].

### 2.2. Role of Mice in the Study of Tuberculosis Development

Working with human lung samples to establish the events that happen from the initial *M. tuberculosis* infection to the development of active pulmonary tuberculosis is complicated for several reasons. Obtaining human lung biopsies is a very invasive approach with a lot of ethical considerations [54], which limits the availability of profitable samples to study. In addition, when it is possible to access these kinds of samples, they usually are either from patients who have been receiving antibiotic treatment, interfering then with the natural course of the infection, or from necropsies. In both cases, it is only the final part of the infection that is being observed, ignoring the events prior to this last phase. Furthermore, working with human lung samples also has the limitation of the limited control of other variables of the infection, such as the mycobacterial strain causing the infection, the infective dose, the time of disease development from the initial infection or factors ligated to the host, such as age, the presence of comorbidities or their environmental and nutritional conditions.

Like for human assays, biomedical research with mice (and with other animal models) needs to be performed under the supervision of the corresponding experimental animal ethical committees. The main objective of the evaluating committees is to guarantee that each project involving animal use follows the three R’s rule: reduce, replace and refine. “Reduce” refers to use the minimum number of animals possible; “Replace” refers to finding new models that can substitute the current in vivo models (e.g., in silico models, organoids, etc.); and “Refine” refers to the search for techniques and experimental procedures that minimize the pain and suffering of the animals, while maintaining the quality of the results and conclusions obtained from those procedures. The laws and the committee’s criteria for evaluating projects can change between countries and continents. For instance, in the United States, the use of laboratory animals is regulated by the Animal Welfare Act, a federal law enacted in the 1960s. This law obligates the institutions that use animals for research to make a yearly count of used animals, classifying them in different categories according to the pain and distress caused to them. However, under this law, mice and rats are not considered laboratory animals. This means that it is very difficult to determine how many of these animals have been used in biomedical science in the US [55]. In Europe, the definitions of “animal” and “procedure” are wider than the ones for the US, so more species are included in the list of regulated animals in Europe compared to the US [56].

One of the strengths of using mice for the study of tuberculosis lies in the fact that researchers can obtain lung samples from, potentially, every timepoint of the infection, allowing them to examine the different stages and phases of the disease. In addition, the same variables that are uncontrolled for human studies are totally controlled in the case of animal models. To this must be added two more aspects: one is the notable degree of similarity between mouse and human, in terms of pulmonary anatomy and immune mechanisms [57]; the other is the susceptibility of mice to *M. tuberculosis* infection [58]. That is why most of the stages of tuberculosis can be shown in mouse models.

By contrast, one of the major flaws of the majority of murine models of tuberculosis is the lack of intragranulomatous necrosis in the lesions, which has led to the concept that, in fact, they behave as tolerant hosts, in contraposition to guinea pigs, able to naturally induce intragranulomatous necrosis [59]. This has been a matter of lots of controversy and the need for trying to understand the nature of this necrosis. 

There are different mice lineages for the study of tuberculosis, each one with its own distinctive genetic background. Generally speaking, mice lineages can be divided in two big groups: “resistant” to TB infection and “susceptible” to TB infection. It should be clarified that mice are considered as “tolerant” hosts for the infection, as they are able to deal with elevated mycobacterial burden without compromising their survival [60], but it is true that certain lineages of mice present different resistance levels to the infection.

Susceptibility to the infection is commonly measured by registering survival data, the pulmonary histopathology and pulmonary colony forming units (CFUs) in loads of mice infected with *M. tuberculosis* and comparing them with survival, histopathology and CFUs data of other mice lineages. This parameter depends on the age and sex of the infected mice, the quantity of administered bacteria, the virulence of the used strain, the route of infection, the genetic background of the host (i.e., the mice lineage that has been used in the assays) and the presence of immunocompromising elements in the host, such as malnutrition, diabetes or co-infections [61,62] (Figure 2). The study of all these traits about susceptibility in mice can provide useful information about *M. tuberculosis* infection and TB development and should be considered in both designing an experiment with mice and interpreting the results of that experiment. Additionally, depending on the aim of the experiment, more assays can be added to it. For example, when the efficacy of a vaccine is evaluated, it should be necessary to add a cytokine profile analysis from the lung, spleen and blood of the animals. The effect of these traits in tuberculosis development using mice is summarized in Table 1.

#### 2.2.1. Age

Although tuberculosis can affect all age groups, children and elderly people are the groups with more risk of developing the disease, the former due to undeveloped immune system and the latter due to age-related biological changes (e.g., appearance of immune-disturbing comorbidities, use of immunosuppressive treatments, the decline of immune system effectiveness, etc.), something that is applicable for other pulmonary infectious diseases [63,64,65]. 

Due to bioethical and physiological reasons, the use of mice younger than 3 weeks for infectious disease experiments is not recommended, because, during that time, the newborn mice depend absolutely on their mother’s care (nutrition and heat, mainly) without whom, the survival chances of the pups are drastically reduced [66]. So, the most available information regarding this parameter is obtained after the comparison between old and juvenile mice.

The potential relation between age and other bacterial pulmonary infections has been shown already in mice, which reinforces the possibility that it also occurs with tuberculosis. In one study, researchers infected mice 6−7 weeks old and mice 24 months old with *Streptococcus pneumoniae* and observed that old mice produced a more aggressive pulmonary inflammatory immune response, characterized in this case by a significantly high neutrophil recruitment to the lung in the acute phase of the infection compared with young mice, although the long-term effects of this difference in inflammation are not shown [67]. Another comparing study between young and old mice infected with *S. pneumoniae* showed, again, more inflammation and neutrophil recruitment in response to the bacterial infection in old animals, which was related to a delayed clearance on *S. pneumoniae* from the upper respiratory tract [68].

Focusing on the effect of age upon tuberculosis in mice, there is a certain level of controversy in the literature. Some authors report that age is irrelevant for controlling the infection. In one project, researchers infected 3- and 24-month-old mice with *M. tuberculosis* and registered bacterial burden from the lungs and spleen of the animals and immunologic mRNA profile. The study showed no significant differences in terms of bacillary load in the organs, although it must be said that old mice presented more disseminated bacteria that the young ones and reduced lymphocyte and cytokine mRNA copies [69].

Other groups relate aging in mice with early resistance to the infection. In two experiments from the same university, researchers infected mice of different months of age with *M. tuberculosis* and compared bacillary load in the lungs corresponding to the earliest phase of the infection, which was defined as between 1 to 35 days post-infection. In both experiences, it was shown that older mice kept lower CFUs during this early stage. It was associated with the presence of higher numbers of unspecific IFN-γ producing CD8 T cells in the older mice [70,71].

Nevertheless, there are studies that relate age with susceptibility. One study performed with mice of almost the same range of ages that the studies commented on in the previous paragraph showed older mice being the most susceptible to *M. tuberculosis* infection. According to the researchers, old mice had inefficient T cell populations, which derived in a poor bacterial clearance [72]. Another group correlated the characteristics and composition of the alveolar fluid of elder mice with worse control of the infection by alveolar macrophages [73]. That study was complemented with other assay, which compared AM responses in front of the infection from old and young mice, showing that old mice AM populations are less able to control the infection [74].

The discordances between the results of each study can be used as an example to see why susceptibility to the infection does not depend on only one parameter. The lineage and sex of mice used were different between experiments. The route of infection was not also the same for each experiment. Therefore, it could be hypothesized that the differences in methodology for each assay have had unsimilar impact on the results and conclusions of the experiments. 

Still, these experiments could serve to remark the importance of T-Cell mediated-immune response in controlling the infection and avoiding or delaying the development of tuberculosis.

#### 2.2.2. Sex

There are some evidences that may indicate a possible relationship between sex and mortality upon hostile environmental circumstances. Registered data from modern age episodes of famine and epidemics shows that women seem to cope better with these critical situations, having higher life expectancy than men in the same scenarios [75]. This is something that could be associated, in the case of infections, to compensatory effects in immunological traits in females, as they invest more energy in reproductive tasks than males [76,77]. 

Sexual differences in susceptibility to mycobacterial infections, including *M. tuberculosis*, is not an exception [78]. According to this, the last global tuberculosis report by the World Health Organization shows that the incidence of cases is higher in men than women in all age ranges (except for kids) and in all geographic areas [79], something that was already reported in tuberculosis reports from previous years. Other studies made with human data also support this observation [80,81,82]. Several reasons have been proposed to explain this trend. The main are sexual differences in steroid hormones, genetics and socioeconomic aspects [83,84].

Focusing on steroid hormones, these molecules can be classified in two groups according to where are produced: the steroids produced exclusively by the adrenal cortex of the adrenal gland, which are mineralocorticoids and glucocorticoids, and the steroids produced by the adrenal cortex, the gonads or the placenta, which are androgens, estrogens and progestins [85]. Steroid hormones have the capacity to interfere with the immune system and disrupt the balance among Th responses, which could potentially lead to the enhancement of tuberculosis progression. In addition, there are evidences of the existence of sexual dimorphism in terms of steroid production [86,87,88,89,90]. 

Genetic differences between male and female are the other intrinsic biological reason that may explain the difference in susceptibility. The X chromosome codifies several immune related genes, some of them actively expressed in the immune response against tuberculosis, and the fact that males only have one copy of that chromosome can also be related to their major susceptibility [91,92]. However, it is necessary to take into account that environmental and epigenetic factors can influence other chromosomes apart from the X.

Sex differences in susceptibility to mycobacterial infections have been already shown in mice. An assay with *M. intracellulare* showed increased susceptibility of male mice to the infection, in terms of higher CFU counts from the lung, liver and spleen and reduced phagocytic activity of the male peritoneal macrophages compared with female CFUs and macrophage activity [93]. Another study from the same group showed the same results for *M. marinum* infection in different mice lineages, pointing also to testosterone as a possible steroid hormone involved in mycobacterial infection susceptibility [94]. Finally, experiments using *M. tuberculosis* infected mice also showed increased susceptibility in males [95], which was attributed to the formation of smaller B cell follicles in the pulmonary lesions caused by the bacteria in males compared to females [96].

The first studies about the relation between steroids and tuberculosis development were performed in the decades of the 1950s and 1960s. In one experiment, researchers administered cortisone (glucocorticoid) to vaccinated and unvaccinated *M. tuberculosis* infected mice. They used two strains of *M. tuberculosis*: one highly virulent and the other less virulent for the mice. All the groups of mice that received cortisone registered worse survival data that the mice without cortisone, even in the vaccinated groups [97]. Another study using cortisone and other glucocorticoids reported similar results [98]. 

On the other hand, another study showed that administration of dehydroepiandrosterone or androstenediol, both steroid hormones but with anti-glucocorticoid properties, enhanced the Th1 response in infected mice, allowing an earlier control of the infection [99]. However, it must be said, none of the previous studies were focused on seeing how the steroid dimorphism could have influenced the outcome of the infection.

In fact, very few studies with mice have assessed the relationship between sex differences in steroid hormones and their impact on tuberculosis development. One experiment was centered on the effect of testosterone (an androgen). For that purpose, they used male and female, both castrated and not-castrated, infected mice and compared the course of infection between groups. Not-castrated males showed worse survival and infection control (CFUs, inflammatory response and granuloma size) results compared to castrated males, females and castrated females, suggesting that testosterone may influence the modulation of immune response against *M. tuberculosis* [100]. Studies about the role of estrogens and progestins in tuberculosis development in mice still need to be performed.

For the study of genetic-related tuberculosis susceptibility factors, what is more commonly performed nowadays is performing whole genome scanning, looking for gene candidates for susceptibility to the disease between different mice lineages [101]. Again, few projects were dedicated to comparing between possible differences according to sex. In one study, researchers could observe that body weight loss after infection was related with different loci in males and females. Interestingly, they also reported longer survival times in males instead of in females [102].

Another way to study the impact of genetics and sex in susceptibility can be conducted following an inverse route, that is, starting from a known gene, integrating it in the mice genome and then studying how the infection develops. In this regard, one experiment was performed using transgenic mice expressing human toll-like receptor 8 (*TLR8*), as it has been shown that polymorphisms in this gene could possibly be related to more susceptibility to *M. tuberculosis* infection. Males from wild type and transgene groups were more susceptible to the infection compared with their respective female groups, showing marked differences in pulmonary CFUs and histological pathology, although the project was not focused mainly on that aspect [103].

Two conclusions can be extracted from this subsubsection: a deeper study of the steroid influence in tuberculosis development should be performed, not only for revealing the interaction between distinct types of this molecule and the disease, but also to see how the sexual dimorphism plays in this aspect; it would also be profitable to conduct a deeper study of how the sexual differences in genetics are related to susceptibility, in order to identify possible new biomarkers for, especially, prognosis of the disease. Nowadays, it is strongly recommended by the ethical committees to include the same number of male and female mice in the experiments, in order to analyze all the data obtained in an assay from each sex.

#### 2.2.3. Route of Infection, Quantity and Virulence of Administered Bacteria

The quantity of administered bacteria depends almost exclusively on the chosen route of infection. Typically, for tuberculosis infection in mice there are four routes of infection: aerosol generation and exposition, intravenous administration, intranasal administration and subcutaneous administration. The first two are the most commonly used.

Aerosol infection is the most similar way to replicate the initial step of *M. tuberculosis* infection in humans. This route of administration has routinely been considered for delivering small numbers of CFUs (around 50−100) to the mice lung, in order to establish chronic pulmonary infection in the animals, although some authors consider that dose quite elevated and not corresponding to the real number of bacteria that arrives to the human lung [104].

By the intravenous administration is the other common route of infection in mice. In this case, administered doses are higher (10^4^−10^5^ CFUs suspended in a volume of 0.2 mL of PBS) in comparison with the aerosol route and it is performed by injecting the inoculum in one of the lateral tail veins [105]. This way of infection leads to the dissemination of the bacteria through all the body of the animal, but primordially the infection is established in the lungs. 

Like in the previous subsubsection about age, there is disagreement about with which route of infection mice are more susceptible. Some authors consider that the aerosol route is more virulent due to the fact that, in this route, the immune response triggered by *M. tuberculosis* is more delayed than in the case of intravenous infection, where the bacteria arrive earlier to the spleen and lymphatic nodules and allow an accelerated development of immunity [106,107,108]. It also has been stated that bacterial growth after aerosol infection in mice resembles more of the epidemiological behavior of tuberculosis in humans, while intravenous infection can be more useful for strain virulence study [109]. 

However, it also has been shown that intravenous infection can lead to a rapid development of tuberculosis reflected in a very marked decrease in the animal survival after 28 days of the infection [110]. It is necessary to mention that, in this last experiment, two mice lineages were used, with only one of them being significantly affected after 28 days, while the other showed more resistance to the infection. That is because the genetic background of mice plays, in fact, a very important role in susceptibility against the infection, which will be developed in the next subsubsection. 

Regarding the virulence of the *M. tuberculosis* strain, it is obvious that the bacteria also play its part in the course of the disease. The continuous co-evolution with different hosts in different environments during thousands of years has caused the appearance of various strains and lineages, with variable degrees of virulence between them [111]. In addition, the commonly used *M. tuberculosis* strains for animal experiments have suffered a long progress of adaptation to the laboratory environment, what has reduced their virulence in comparison to clinical isolates [112], something that must be considered in efficacy assays of new treatments and vaccines.

Virulence is a difficult concept to define due to the fact that it not only depends on the side of the pathogen, but also on the side of the host and on the interactions between the former and the latter [113]. We can define virulence as the capacity of a pathogen to infect, replicate and cause disease and/or death in a susceptible host. Using mice to evaluate the virulence of a particular *M. tuberculosis* strain can provide useful information about genomic and epidemiologic data in tuberculosis outbreaks [114] that can be compared to data from previous episodes of tuberculosis. Furthermore, virulence experiments in mice are also performed to measure the virulence of new tuberculosis vaccine candidates, specifically live-attenuated vaccines [115]. Challenging mice with different strains of *M. tuberculosis* also allows for linkage of the genomic differences and virulence factors between bacterial strains with susceptibility to the infection [116,117]. 

#### 2.2.4. Genetic Background (Lineages of Mice)

This is, together with bacterial virulence, the most determinant factor of susceptibility to *M. tuberculosis* infection. Mice lineages can be classified according to its haplotype regarding the type of major histocompatibility complex (MHC), which, in mice, is called H-2 complex [118]. In addition, considering that mice are mainly tolerant hosts, we can also differentiate lineages on its degree of resistance against the infection. Resistant mice take longer than susceptible mice to develop the disease, defined on the bacillary concentration and survival, but, in the end, they all die because of the infection, although some of them survive the same amount of time as non-infected mice. So, the phenotypic difference between the two groups is the time to death and the bacillary load, and the cause of that difference lies in the genetics of each lineage [119,120], which can possibly have an effect on the immunopathology of every type of lineage [121]. Interestingly, only haplotype H-2^k^ has been associated with the evolution towards active TB, as in the other haplotypes, they develop a sort of chronic infection, showing a tolerant phenotype with a different resistance degree among the lineages [122]. In this review, we have included the most used mice lineages for tuberculosis experiments: C3HeB/FeJ, DBA/2, 129/Sv, BALB/C and C57BL/6.

##### C3HeB/FeJ

This lineage develops active TB, showing a non-tolerant, susceptible phenotype. The main characteristic of this model is its ability to develop well-formed granulomas with central necrosis and liquefaction [123]. It is the only mice lineage capable of generating human-like lesions of active tuberculosis, so it makes it the predilect model for researchers to study the active tuberculosis pathology. Although this model can be infected also by aerosols, this route causes an unsimilar impact in each animal of the experiment, with the appearance of different types of pulmonary lesions, each one with its own cellular composition and bacterial burden [124]. Additionally, the necrotic lesions of this model are hypoxic [125], similar to human active tuberculosis necrotic lesions. This fact is of great importance attending to the capacity of *M. tuberculosis* to activate intrinsic genetic factors related to reduction of metabolic activity, survival against hostile environments and the host’s immune system and, potentially, acquisition of antibiotic resistances under hypoxic conditions [126,127,128].

The source of the high susceptibility of the lineage resides in its genome. The group that firstly described this mouse identified polymorphisms in locus *sst1* that were responsible for enhanced effects on lung mycobacterial burden and necrosis control of infected macrophages, as part of a multigenic network that controls susceptibility to *M. tuberculosis* [129]. Further analysis revealed that the expression levels of *ipr1*, a gene related to resistance to the infection and encoding inside *sst1*, are severely reduced in this mice lineage, in comparison to a resistant mice lineage [130]. How this gene is related to resistance/susceptibility to infection is something not already known, but it could possibly be an enhancer of antibacterial functions of macrophages [131]. Another locus related with susceptibility has been identified in chromosome 7. This locus seems to have a systemic effect on the host in later stages of the disease but acting as well on macrophages. It also belongs to the same multigenic network as *ipr1* [132].

Attempts to apply this knowledge to human tuberculosis have shown that *SP110*, the homolog of *ipr1*, could be a good marker of susceptibility but only for some populations [133,134,135]. There is no study about chromosome 7 locus and tuberculosis in humans.

With this lineage, it has been seen how neutrophils are the principal cellular component responsible for the excessive inflammatory response involved in the development of active tuberculosis, mainly due to the formation of neutrophil extracellular traps that enhance extracellular bacterial growth [136,137]. This fact prompted the development of host-directed therapies based on reducing inflammation, using this model for the initial tests of new compounds or strategies [138,139]. Interestingly, although it shares the haplotype k with the C3H/HeN lineage, the latter does not develop as much pulmonary damage as the C3HeB/FeJ lineage, probably due to differences in the inflammatory response against the infection between lineages [136].

These mice have also been useful for studying pathogenic traits of *M. tuberculosis*. In one study, our group remarked the cording phenomenon, i.e., the aggrupation of mycobacterial cells in long aggregates called “cords” [140], as a key virulence factor for *M. tuberculosis* to induce active tuberculosis after intravenous administration [141], which is something that should be considered in every assay involving this mouse lineage, *M. tuberculosis* and intravenous infection. Another group studied the genetic signature of *M. tuberculosis*, using these mice to as an in vivo environment and registered an extensive list of genes and pathways associated to the in vivo requirements of the bacteria to develop the active disease [142].

##### DBA/2

This is another susceptible lineage for *M. tuberculosis* infection, characterized by a faster progression of lesions in the lung due to a major bronchogenic dissemination of infected foamy macrophages [143]. It belongs to the haplotype d. Interestingly enough, the intranasal inoculation of LPS was not enough for developing intragranulomatous necrosis [144]. 

In this lineage, there are also genetic traits possibly associated with susceptibility to the mycobacterial infection. Researchers found that the genetic construction of two loci in chromosomes 3 and 19, *TRL3* and *TRL4* respectively, of this lineage genome were related to increased bacterial burden in comparison to a resistant lineage of mice [145]. Unlike the case of C3HeB/FeJ and *ipr1*, the implication of these loci in human susceptibility to tuberculosis has not been studied already.

On non-direct genetic related susceptibility traits, one study found that bone marrow derived macrophages of DBA/2 mice were significantly less able to control *M. tuberculosis* infection, in comparison to other macrophages isolated from resistant lineages [146], highlighting the importance of initial innate immune response, especially macrophages, in controlling the progression of the disease. Another study showed that, upon infection, this lineage presents a reduced number of pulmonary mucosal dendritic cells and regulatory T cells, again compared with resistant lineages [147]. The genetical mechanisms responsible of these traits (if there are any) remain unknown. 

Several genes from *M. tuberculosis* that are crucial for its pathogenesis have been identified using this lineage of mice. For instance, in one study the virulence of three *M. tuberculosis* mutants of genes encoding for methionine, proline and tryptophan synthesis were measured and compared to wild type *M. tuberculosis*. Mutants in proline and tryptophan genes were classified as avirulent. Researchers also compared these mutants with BCG in a protection assay as attenuated vaccine candidates. In that assay, mutants showed better protection efficacy [148]. Another group showed that a *M. tuberculosis* mutant in genes encoding mycolic acid methyltransferases related to modifications of the lipids of their cell wall [149] presented lower virulence than wild type *M. tuberculosis*. It also was tested as a possible vaccine candidate but using other mice lineages [150].

##### 129/Sv

It is classified as susceptible, belonging to the haplotype b group, even if is not capable of generating intragranulomatous necrotic lesions such as C3HeB/FeJ does. At first, it was thought that one of the causes of its susceptibility, as well as for the rest of susceptible lineages, was related to *Nramp1* gene, which had been associated previously to resistance to other intracellular pathogens. However, works with mutants of 129/Sv lacking *Nramp1* and other lineages showed that, apparently, this gene has no effect on resistance to *M. tuberculosis* infection [122]. Another assay was focused on the possible role of mycobacterial *Nramp1*, homologous to the mammalian *Nramp1*, but it was shown that this gene was not involved in the virulence of the bacteria [151]. However, it seems that the Nramp1 protein contributes somehow to the functionality of the macrophage phagosome [152]. Signaling by Dectin-1 also does not seem to affect resistance or susceptibility of this lineage to tuberculosis [153].

Nitric oxide is produced by macrophages after being activated with IFN-γ in order to deal with such intracellular pathogens as *Listeria monocytogenes* or *M. avium* [154] and also against *M. tuberculosis* in both human and murine environment [155,156]. Focusing on this mice lineage, in one study, researchers used hybrid mice with the genetic background of 129/Sv and C57BL6 (noted below) that carried a disruptive mutation in *NOS2* gene, which encodes an inducible nitric oxidase synthase that participates in the production of reactive nitrogen species (RNS), to measure the impact of this gene in tuberculosis resistance. They arrived at the conclusion that this gene is necessary to establish a bacteriostatic control of the infection [157].

In fact, hybrid mice combining 129/Sv and C57BL6 genomes have been widely used for particular mutant mice obtention and the implication of that mutation in tuberculosis studies. For instance, in one study, these two mice lineages were used for obtaining mice defective in TNF-α and IFN-γ and to study the implication of these two molecules in the formation of granuloma upon *M. tuberculosis* infection and control of the infection [158]. In another assay, these hybrid mice were used to obtain mutants deficient in the production of IL-6 to study its role in tuberculosis control. Researchers found this molecule totally essential to contain the infection [159].

Apart from the previous examples, the contribution of TAP1, which is a transporter associated with antigen processing and presentation to CD8 T-Cells, was shown using this method of combination between 129/Sv and B6 mice. Mutant mice unable to produce TAP1 presented higher mortality and bacillary load in lungs, spleen and liver, compared to non-mutant mice, remarking the importance of CD8 T cells’ action in tuberculosis control [160]. Another study showed that TIR8 is necessary to control and regulate the inflammatory response, and its absence leads to higher tuberculosis susceptibility [161]. 

However, there are also assays comparing physiological aspects of the two lineages. It was found that 129/Sv mice under *M. tuberculosis* infection express more acetyl-CoA carboxylase 2 at the macrophage level than C57BL6 macrophages. This enzyme is related to the formation of foamy macrophages through cellular lipid accumulation, and, apparently, this overproduction can be triggered and profited by *M. tuberculosis* in order to progress with its infection [162], something that might be related to the fact that 129/Sv mice absorb more cholesterol and fats that C57BL6 mice with the same diet [163]. 

Following with macrophage-related pathogenicity upon infection in 129/Sv, histone deacetylases appear to be involved in some regulatory functions that might benefit mycobacteria. One of the groups that has been associated with immunomodulatory properties is the family of sirtuins, especially sirtuin 1 [164]. Using 129/Sv macrophages it was found that sirtuin 1 is able to avoid macrophage apoptosis to improve the control of *M. tuberculosis* infection [165]. In addition, another assay not only confirmed the importance of sirtuin 1 in this model, but also showed that this molecule is reduced in human cases of active tuberculosis compared to healthy people, making Sirtuin 1 an interesting possible new biomarker of tuberculosis in humans [166]. 

##### BALB/c

This lineage has been classified as resistant to the infection, although other authors consider the lineage as susceptible. It shares a haplotype with the DBA/2 lineage. Nevertheless, this model has a better control of Mtb infection than the previous ones. The infection course is divided into an acute and a chronic phase, each one with a distinct histologic and immunologic profile [167]. This has been the “gold standard” lineage used to test chemotherapy against TB [168], although it was suggested that C57BL6 could behave as BALB/c in terms of chemotherapy trials [169]. 

Some authors consider that the characteristics of the infection in these mice are already representative of the latent infection. But others have implemented additional steps to achieve a higher parallelism with the human latent infection, modifying the “Cornell Model” of latent infection originally developed in outbred Swiss mice [170]. These steps are usually: (1) a short-period antibiotic treatment after infection to reduce the initial bacterial burden; (2) a pre-immunization with BCG before infection and the short-period antibiotic treatment after the infection [171,172]. On the other hand, other groups use auxotrophic mutants that can grow and establish in the lungs in the presence of the auxotrophic compound (e.g., streptomycin), but when it is depleted, the bacteria adopt an inactive state [173]. In any case, the utility of a latent model is to test new drugs and treatments to eliminate the latent bacilli or to develop new technics for diagnosis.

It is in the chronic phase when more pulmonary damage is produced, and this is associated with an exacerbated IL-4 production by Th2 cells, which breaks the necessary Th1/Th2 balance to control the infection [174]. In fact, using BALB/C mice with unfunctional IL-4, researchers reported a possible toxic relation between IL-4 and TNF-α in the chronic stage of the disease [175]. However, in a previous assay with IL-4 deficient BALB/C, another group of researchers showed the opposite effect, i.e., the lack of IL-4 aggravates *M. tuberculosis* infection [176], which reinforces the needed of balance between Th1 and Th2 responses in tuberculosis. In humans, IL-4 may also have a determinant role in tuberculosis development [177]. 

Another cytokine that could possibly participate in the response against the mycobacteria is IL-12, as this molecule is a strong Th1 inducer cytokine [178]. One group tested in this and in other mice lineages the effect of administered IL-12 on the *M. tuberculosis* infection. According to the authors, the administration of IL-12 increased mice resistance to the infection by a possible initial increment in lymphocytic IFN-γ production and by downregulating the expression of IL-4 [179]. Interestingly, another group showed how *M. tuberculosis* is able to modulate the production of IL-12 in bone marrow derived macrophages from BALB/C mice [180]. High levels of IL-12 have been reported in human tuberculosis patients. This elevated production could be an attempt to restore the Th1/Th2 balance disturbed by the also elevated production of IL-4 in these patients [181,182]. 

Recently, focusing on the bacterial side, a study has been published where researchers use this mice lineage to characterize the expressed genomic profile of *M. tuberculosis* in the initial stages of infection and pulmonary establishment. From a long list of genes, the group identified *rv0180* as a new possible virulence factor of *M. tuberculosis* in mice infection. It also seems to be involved in human macrophage infection [183].

Attenuated *M. tuberculosis* mutants have been tested also in this lineage as possible new vaccine candidates. After showing reduced virulence in mice, an *M. tuberculosis* mutant in *sigE*, which encodes a structural unit of helper factors of the mycobacterial RNA polymerase, was compared to BCG against two *M. tuberculosis* strains. The mutant showed considerably better protection results than BCG [184,185].

##### C57BL/6

It is also known as B6, and it belongs to the group that is resistant to infection. This lineage develops severe pulmonary damage after a long period since infection, around 200 days. One important concern about this lineage (on most occasions) is that is not able to produce marked necrotic lesions such as C3HeB/FeJ when the disease is highly advanced [186], although this, apparently, is highly dependent on the *M. tuberculosis* strain used in the experiment [187]. The absence of NOS2 also has been associated with the production of necrotic lesions in these mice [188], something that marks the importance of RNS, in contrast with reactive oxygen species (ROS), which seems to have a minor role in the progression of the disease [189].

Like BALB/c, due to the slow progression of mycobacterial infection, this model offers the opportunity to recreate the latent tuberculosis environment. In this case, to establish the latent infection, the animals are infected via low-dose aerosol and treated with antibiotics for several weeks [190]. In this lineage, the strategy of auxotrophic mutants of *M. tuberculosis* to achieve the latent state has also been used [191]. Another group used intradermic administration in the ear to stablish the infection in the local ear lymph node, from where, under certain circumstances, it can progress systematically [192]. 

Again, certain cytokines may contribute to that process. One group worked with transgenic B6 mice overexpressing IL-10. They showed, after *M. tuberculosis* infection, that IL-10 acts in late stages of the infection. Compared to wild type B6, transgenic mice had superior pulmonary mycobacterial burden, reduced levels of TNF-α and IL-12p40 (a component of IL-12) mRNA and reduced capacity of lung cells to produce IFN-γ [193]. Regarding IL-12 role, another study showed again that this interleukin is necessary to induce the production of IFN-γ and to establish a competent response against *M. tuberculosis* [194] and another showed that IL-6 is necessary also to induce the production of IFN-γ [195].

Surprisingly, another group reported that deficient T-bet B6 mice, which were not able to assemble a proper Th1, had increased levels of IL-10 during all the infection that could have contributed to the augmented susceptibility of those mice compared to wild type B6 [196]. Moreover, another group showed that both B6 and BALB/c mutant lineages deficient in IL-10 production presented better control of the infection, associated with an early development of Th1 response [197]. The role of IL-10 in human tuberculosis, as well as in other infectious diseases and mice models, is reviewed more deeply elsewhere [198,199].

Recently, researchers have pointed to microRNAs (miRNAs) as molecules of interest in the study in tuberculosis development [200]. The B6 lineage has been largely used for that purpose. For instance, in one study, *miR-223* was identified as a key factor of susceptibility to the infection. According to the authors, the depletion of *miR-223* from B6 genome turned the resistant lineage completely susceptible. Apparently, it has an essential role in organization of inflammatory response in the lung against *M. tuberculosis* infection [201]. Another miRNA involved in susceptibility is *miR-155*, in charge of increasing the Th1 response in the form of regulation of IFN-γ secreting CD4 T cells, without which B6 mice became more susceptible [202]. *miRNA-223* is upregulated in human active tuberculosis cases and inhibits apoptosis of macrophages [203], while some authors have suggested *miRNA-155* as a possible biomarker for diagnosis of active tuberculosis, especially in children [204].

Other cellular processes for controlling the mycobacteria seem to be also influenced by miRNAs in mice. Autophagy regulation has been associated to *miRNA-27a*, *miRNA-33* and *miRNA-125a*, each molecule intervening in different pathways [205,206,207]; apoptosis and inflammatory response are controlled by *miRNA-27b* [208] or glycolysis and IL-1β production mediated by *miRNA-21* [209]; these are only a few examples of a very long list of miRNAs involved in tuberculosis containment [210].

Resuming the concept of the therapeutic vaccines against tuberculosis, there is a big concern since Robert Koch’s tuberculin tests in people infected with *M. tuberculosis*: the fear of triggering the exacerbation of tuberculous foci in infected people, such as what happened in the case of tuberculin and Koch’s assays [211,212]. This is something of particular importance in the case of latent tuberculosis individuals, in whom this possible toxic effect could make the disease progress to an active form. 

In fact, one of the main objectives of this kind of vaccine, according to the World Health Organization, should be to prevent the development of active tuberculosis from latent infection. Moreover, it is also desirable that therapeutic vaccines could shorten the duration of the actual treatments [213]. Therefore, B6 mice offer a good opportunity to test if a therapeutic vaccine is able to achieve both objectives [214,215]. The main characteristics of this and the other mice lineages are represented in Figure 3.

#### 2.2.5. Immune Imbalance Scenarios

There are two types of situations where the immune system of the host can be unbalanced or compromised: an infectious scenario in which the immune system has to deal with a determined pathogen, and a non-infectious scenario, such as allergies or diabetes, where there is an intrinsic dysregulation of the immune system without the appearance of an external factor.

In the case of the infectious situation, and relating it with tuberculosis susceptibility, an already existing infection could facilitate the co-infection by *M. tuberculosis*. The clearest example of this could be the infection with human immunodeficiency virus (HIV). As HIV targets CD4 T cells and depletes this cellular population, the infected host becomes susceptible to infection by other pathogens. Mice have been used to study the outcomes of *M. tuberculosis*–HIV coinfection, showing how the viral infection affects several aspects of immunology, pathology and treatment of the mycobacterial infection [216,217,218].

Other viral infections have also been related to the promotion of tuberculosis development, not by downregulating the immune response such as HIV, but for increasing it, especially for inflammation. The most common inflammation-associated respiratory virus has always been influenza virus, and studies about co-infection with *M. tuberculosis* point out that the virus enhances the pathogenicity of the bacterial infection [219]. 

The recently appeared SARS-CoV-2 is another example of a virus that causes exacerbated pulmonary inflammation. However, in this case, there are very few projects with mice that are focused on its effect upon *M. tuberculosis* infection. In one of them, researchers used a B6 model of latent tuberculosis and infected it with murine hepatitis virus (another coronavirus). They reported increased mycobacterial burdens in the organs of animals co-infected in comparison to animals with only mycobacterial infection [220]. On the other hand, in another assay using a humanized B6 lineage [221], it was reported that SARS-CoV-2 infection did not enhance the mycobacterial progression [222]. Nevertheless, a third group using the same humanized model, although they did not register increased pulmonary damage, found increased bacterial burden and augmented expression of IL-13 in animals coinfected with *M. tuberculosis* and SARS-CoV-2 [223]. What is interesting about this last finding is that overexpression of IL-13 has been linked to increased tuberculosis susceptibility in a B6 mice model [224], while, in humans, some polymorphisms of IL-13 genes have been associated also with increased susceptibility to the disease [225].

Focusing now on parasitic infections, it is interesting to note the overlap in geographical distribution between gastrointestinal parasitic infections and tuberculosis. Current experimental results reveal certain controversy about the effect of intestinal helminthiasis on tuberculosis development. The immune response displayed against parasitic infections is mainly Th2 [226], so it is reasonable to consider that this could augment the susceptibility to tuberculosis. While some researchers report that this Th2 response is responsible for enhancing tuberculosis progression in mice co-infected with helminths and *M. tuberculosis* [227,228], others found that pre-existing helminthiasis had no effect or even had a protective effect [229,230]. The lack of homogeneity between experimental conditions between assays (mice lineages, parasitic species, etc.) might explain the discrepancy found. 

Another parasitic disease present in areas with high prevalence of tuberculosis is malaria. Malaria is caused by several species of the *Plasmodium* genre, each one of them with a complex life and development cycle. Traditionally, it was considered that the immune response against the parasite was only based in balance between Th1 and Th2 cell subsets, but, recently, researchers showed that more types of immune cells might are involved [231]. Regarding the co-infection scenario with *M. tuberculosis*, *Plasmodium* seems to alter the immune response against the bacteria in the same way that intestinal parasites do, that is, augmenting Th2 responses [232]. In fact, this is what has been observed in a mouse model of coinfection. Mice coinfected with both agents showed a shift from Th1 response to Th2 response, which led to reduced survival rates, increased bacterial burdens and exacerbation of pulmonary damage due to tuberculosis [233]. Interestingly, in another experiment using a mouse model of co-infection challenged with a non-lethal strain of *Plasmodoium*, researchers found similar results regarding tuberculosis pathology enhancement, but they also found increased populations of Th1 cells in the lungs. The group suggested that these cells could represent inefficient cells against tuberculosis or cells that were destinated against *Plasmodium* instead of the bacillus [234].

In the case of non-infectious scenarios, two others studied regarding their relationship with tuberculosis are malnutrition and diabetes. Malnutrition can include both sides: undernutrition and overnutrition. In both cases, the dysregulation of the energy intake affects negatively the immune system [235]. It is clear the impact of undernutrition in tuberculosis development, as most of the global zones where undernutrition is present overlap with the zones with higher incidence of the disease [236].

It has been shown, particularly in mice, the effect of protein deficit on tuberculosis infection. Mice that received a poor protein diet succumbed earlier to the infection, which was associated to an impaired production of IFN-γ, TNF-α and NO synthase [237]. It also has been shown that vitamin D has a positive role in tuberculosis containment, so the absence of it from the diet could potentially increase the susceptibility to the infection [238,239].

The effect of overnutrition is not as marked as the effect of undernutrition, but while some studies in humans report that, apparently, obesity is related to enhanced resistance to the infection [240,241], studies with mice point to the opposite direction, finding that obesity models obtained by administration of high fat diets deal worse with *M. tuberculosis* infection and relate this susceptibility difference to the intestinal microbiome change caused by the diet [242,243,244]. So, it can be hypothesized that, in the case of overnutrition, as long as the intestinal microbiome, which depends on other factors than the nutritional income of the diet, remains diverse and functional, there might be no additional enhancement of susceptibility, but other parameters must be included to evaluate the status of the host.

Lastly, diabetes and tuberculosis has been associated since the 1930s [245] and still, nowadays, can be considered risk factors that increase susceptibility [246]. Both types of diabetes are characterized for presenting chronic inflammation [247], being the main point of connection between that disorder and tuberculosis. Regarding this, one group found that diabetic mice had increased levels of a certain natural killer (NK) cell population that produced higher levels of IL-6 and augmented the inflammatory response and the pulmonary tissue damage upon infection [248]. 

Additionally, one group showed that diabetic mice were more susceptible to the infection in terms of bacterial burden and inflammatory histopathological visualization of the lung [249]. Then, they performed a complementary assay where they found that diabetic mice take longer to transport *M. tuberculosis* to the lymph nodes and develop an effective adaptive immune response [250]. In addition, they also found that AMs from diabetic mice were less able to phagocyte *M. tuberculosis* that AMs from non-diabetic mice [251].

## 3. Conclusions

In comparison with the initial experiments using animals for the research of infectious diseases, nowadays, the assays with animals are performed in a much more sophisticated way. Statistics are applied for calculating the optimal number of individuals per group necessary to obtain significant results; the housing and feeding condition are homogeneous for all the animals of a study, as well as the procedures and the analysis of the results, something that was unthinkable 200 years ago, when experimentation on animals was introduced in biomedical research of infectious diseases. This standardization of the experimental conditions has been possible thanks to the big names mentioned above, such as Koch, Pasteur, Abbie Lathrop or Little, among others.

In the field of tuberculosis, research using mice and other rodents has provided useful knowledge about the susceptibility of the host to *M. tuberculosis* infection. That knowledge has been used to know the pathological pathway of the bacteria once it infects a host, as well as to create and perfect tools for diagnosis, treatment and prevention of tuberculosis. The different lineages of mice commented above have been used for the research of those four elements of tuberculosis control, but we can highlight some indications of each lineage. Susceptible lineages, especially C3HeB/FeJ, can be used to study the pathology of active tuberculosis scenario, as well as to develop new therapeutic strategies for this state of the disease, such as the case for host-directed therapies. In addition, to determine the cytokine profile displayed in susceptible mice upon disease might help to establish which human cytokines could be used as biomarkers of diagnosis and progression of the active stage. That would serve not only for diagnosing the disease, but also as a prognosis tool to evaluate the evolution of the treatment in patients with active tuberculosis. Furthermore, resistant lineages should be used to study the latent tuberculosis scenario. In this case, the study of the cytokine profile, or other new molecules such as miRNAs in C57BL/6 mice, could offer new diagnosis tools for human latent tuberculosis. Finally, vaccines can be tested using the two types of lineages, while always being aware of the characteristics of the infection in each type of lineage. However, several comments can be made about some deficits in the study of susceptibility traits. 

First, there are discrepancies regarding the role of some traits upon tuberculosis susceptibility. The clearest example of this can be observed in the case of aging. In certain ways, it also can be seen for the sexual dimorphism, the route of the infection and some of the unbalanced immunity scenarios. It can be hypothesized that the contradictory results can be due to the methodological differences of the experiments: different mice lineage, different range of age, different route of infection and quantity of administered bacteria, etc. As all these factors, separately, have a specific effect on susceptibility, the possible combinations between them can make it so that some overcome the effect of others, changing the expected results. When it comes to designing an experiment, the most important factor is the lineage of mice that is intended for use, as there are significant differences in the lesions that each type of lineage can develop. Additionally, for some of the lineages, the route of infection for obtaining the desired lesions is already established. The best example is C3HeB/FeJ mice and the intravenous administration of *M. tuberculosis*. Only with this route of infection is the formation of liquified pulmonary lesions guaranteed. So, if, for example, the purpose of the experiment is to study the effect of age upon development of active tuberculosis, it should be performed using old and young C3HeB/FeJ mice infected intravenously with *M. tuberculosis*.

Second, there are aspects of some traits that need to be more deeply studied. That is the case of the role of differences in steroid production between sexes. What could be performed is to stimulate the steroid production in mice, confirm the presence of sexual differences in glucocorticoid production and evaluate if these differences have influence in tuberculosis progression, always taking into account the strain of mice and the route of infection. The type of steroid seems to be important also, as, apparently, some types of steroids avoid tuberculosis progression and other types enhance it. 

Research with mice could also be implemented in revealing the role of antibodies in the tuberculosis course, but also in tuberculosis diagnosis and treatment. Currently, new strategies in diagnosis based on detection and quantification of antibodies in blood samples, using them as biomarkers of the disease, as well as therapeutic strategies based on administration of antibodies, are gaining popularity [252]. Proof of concept assays could be performed using the different mice models to explore the antibody patterns and responses generated under *M. tuberculosis* infections, from active stage (C3HeB/FeJ) to latent stage (C57BL/6).

This can also be applied for the study of unbalanced immune scenarios, particularly in the case of pulmonary co-infections. Nowadays, the main pulmonary microbiological threat is SARS-CoV-2. The virus has two hypothetical possible ways to exacerbate the progression of tuberculosis, both related with immune system disfunction: one is by increasing the inflammatory environment of the lung, as the immune response against SARS-CoV-2 is markedly inflammatory. The other way directly interferes with the immune response against tuberculosis, something already seen in humans [253,254]. The humanized mice model noted above offers a good opportunity to study the *M. tuberculosis*-SARS-CoV-2 co-infection scenario. The benefits of this research go beyond this pandemic. The knowledge generated could be used in future pandemics caused by agents that share pathological characteristics with SARS-CoV-2.

On the other hand, there are parameters that have been totally ratified. The degree of susceptibility depending on the lineage of mice used in the experiment has been confirmed by several groups, which has allowed us to determine which specific lineage of mice would be useful for a certain experiment. In addition, there are specified protocols for each lineage to obtain a desired model of active or latent tuberculosis and also protocols for testing new drugs or treatments against the disease. 

In conclusion, great advances have been made since the beginning of the biomedical research of tuberculosis, thanks in great part to the use of the mouse model, as it is a cheap, easy-to-handle and reliable model, but much more is needed in order to improve the control of the disease.

## Figures and Tables

**Figure 1 pathogens-12-00049-f001:**
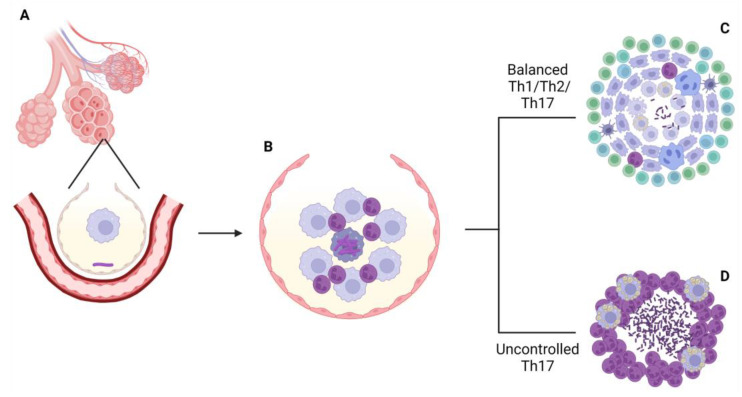
*M. tuberculosis* arrives to the host’s alveoli after inhalation of infected aerosols (**A**). There, the bacillus faces pulmonary surfactant secreted by type II pneumocytes and encounters the alveolar macrophage. Once it’s phagocytized, *M. tuberculosis* secretes ESAT-6 peptide to avoid the formation of phagolysosome and escape to the cytosol of the AM. That allows the bacillus to replicate and kill the AM. This causes the inflammation of the alveoli and the entrance of more AM and neutrophils (**B**). The inflammation leads to the drainage of alveolar fluid to the lymph nodes, where antigen presentation will occur. From there, two possible endings of the infection can happen: if there is a coordinated immune response between Th1, Th2 and Th17 responses, the infection will be controlled (**C**). If there is an excess of Th17 response, there will be excessive neutrophilia, disorganized granuloma formation and tissue destruction, causing the dissemination of the infection (**D**).

**Figure 2 pathogens-12-00049-f002:**
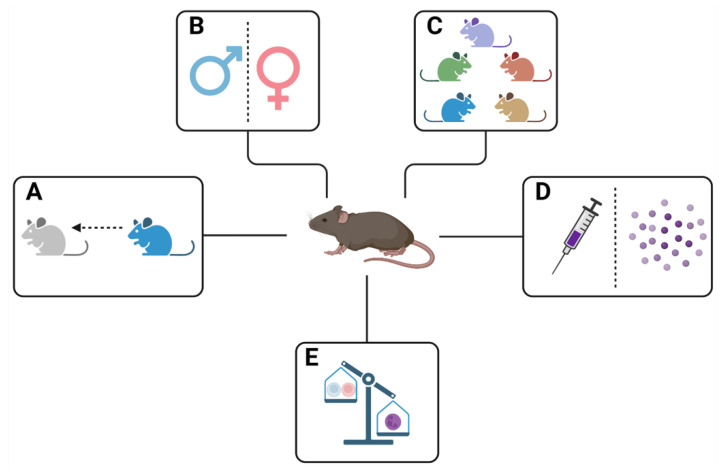
Representation of the traits that can influence development of tuberculosis: (**A**) Aging; (**B**) Differences between sexes; (**C**) Genetic differences between different lineages of mice; (**D**) The route of infection (intravenous or aerosol); (**E**) Immunocompromising elements that can unbalance the immune response against infections (e.g., diabetes, coinfections, etc.).

**Figure 3 pathogens-12-00049-f003:**
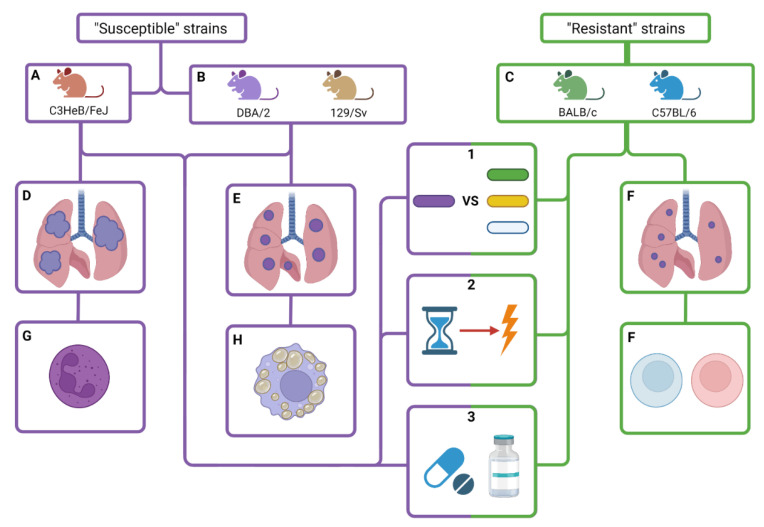
Although being tolerant hosts for *M. tuberculosis* infection, mice can be classified in two groups regarding tuberculosis development. There are “susceptible” mice, which are C3HeB/FeJ (**A**), DBA/2 and 129/Sv (**B**), and there are resistant mice, which are BALB/c and C57BL6 (**C**). C3HeB/FeJ mice are able to develop human-like active tuberculosis liquefied lesions (**D**) due to the hyper-inflammatory response upon the infection leaded, mainly, by neutrophilic infiltration (**G**); DBA/2 and 129/Sv mice are also classified as susceptible, but pulmonary lesions are not necrotic nor liquefied (**E**) and the cause of their susceptibility is the dissemination of infected foamy macrophages (**H**) to other lung areas. On the other hand, resistant mice BALB/c and C57BL/6 develop smaller lesions (**F**) than susceptible mice because the immune response in this case is leaded mainly by lymphocytes (**F**). Both susceptible and resistant types of mice can be used for: (1) Evaluating the virulence of different strains of *M. tuberculosis* and comparing them to more common lineages; (2) Studying the mechanisms that unleash the development of tuberculosis to the initial latent infection; (3) Testing new drugs, vaccines and treatments against *M. tuberculosis*.

**Table 1 pathogens-12-00049-t001:** Main findings relating age, sex, infection route, mice lineage and unbalanced immunity with susceptibility to *M. tuberculosis* infection.

Reference	Susceptibility Trait Analyzed	Aim of Study	Measurements	Main Findings
[69]	Age	Course of the infection in old vs. young mice	-Bacterial burden in lungs and spleen -mRNA from lung tissue	-No differences in bacterial burden between young and old mice -More bacterial dissemination in old mice -Reduced lymphocyte and cytokine mRNA copies in old mice
[70,71]	Age	Course of the infection compared between mice of different ages	-Bacterial burden in lungs -Accumulation of lymphocytes in lungs and spleens -Cytokine production by lymphocytes	-Incremented presence of CD4 and CD8 T cells in the organs of old mice -Increased production of Th1 cytokines by T cells of old mice -Reduced bacterial burden in old mice
[72,73,74]	Age	Course of the infection compared between mice of different ages	-Survival of infected animals -Bacterial burden in lungs, spleen and liver -Lymphocyte’s isolation and characterization -Characterization of alveolar lining fluid (ALF) of old mice -AM’s isolation and characterization upon *M. tuberculosis* infection	-Reduced survival of old animals [72]; no effect on survival [73] -Inability of old mice to clear the infection from the organs -Lack of T cell response in old mice -ALF from old mice promotes the development of tuberculosis -AMs populations from old mice promote the development of tuberculosis
[95,96]	Sex	Course of the infection in males vs. females	-Survival of infected animals -Bacterial burden in lungs, spleen and lymph nodes -Histological analysis of damaged lungs -Pro-inflammatory cytokine production -Lymphoid aggregates formation	-Reduced survival of males -Higher bacillary loads in males -Same pulmonary damaged area between sexes. -Inflammatory cytokines production more elevated in males -Smaller lymphoid aggregates in males
[97,98]	Sex	Steroid effects upon the course of the infection *	-Survival of infected animals -Bacterial burden in lungs and spleen	-Reduced survival of the animals that received cortisone or corticotrophin -High bacillary loads in animals that received cortisone or corticotrophin
[99]	Sex	Steroid effects upon the course of the infection *	-Survival of infected animals -Bacterial burden in lungs and spleen -Cytokine profile expression -Histological analysis of damaged lungs	-Reduced survival in mice with high levels of testosterone -High bacillary load in mice with high levels of testosterone -Reduced expression of pro-inflammatory cytokines in mice with high levels of testosterone -Bigger damaged area in mice with high levels of testosterone
[100]	Sex	Steroid effects upon the course of the infection *	-Survival of infected animals -Bacterial burden in lungs -Histological analysis of damaged lungs -Presence of delayed hypersensitivity (DH) -Cytokine profile expression	-Enhanced survival in mice treated with steroids -Lower bacillary loads in mice treated with steroids -Smaller pulmonary damage in mice treated with steroids -More sustained DH in mice treated with steroids -Augmented Th1 cytokines expression in mice treated with steroids
[102]	Sex	Genetic relation between sex and susceptibility	-Survival of infected animals -Body weight of the animals -Detection of quantitative trait loci (QTL)	-Reduced body weight and survival in females -Males and females presented different QTLs associated with the post-infection outcome
[103]	Sex	Genetic relation between sex and susceptibility	-Bacterial burden in lungs, liver and spleen -Histological analysis of damaged lungs	-Higher bacterial loads in males -Bigger pulmonary damaged area in males
[106,107,108,109]	Infection route	Course of the infection upon aerosol vs. intravenous administration	-Survival of infected animals -Bacterial burden in lungs, liver and spleen -Histological analysis of damaged lungs -Determination of cytokine mRNA profile -Quantification of IFN-γ producing cells -*M. tuberculosis* fitness in the lungs	-Higher bacterial loads in mice infected by aerosols -Reduced survival of mice infected by aerosols -Higher pulmonary damage in mice infected by aerosols -Bigger pulmonary granulomas in mice infected by aerosols -Higher number of IFN-γ producing cells in mice infected intravenously -Earlier IFN-γ production in mice infected intravenously -*M. tuberculosis* fitness after aerosol infection is more realistic that after intravenous infection
[110]	Infection route	Course of the infection upon aerosol vs. intravenous administration	-Survival of infected animals -Bacterial burden in lungs and spleen -Histological analysis of damaged lungs -CD4 and CD8 T cells quantification	-Reduced survival of one of the mice lineages used after intravenous infection -Higher bacterial loads in that same lineage -Reduced number of T cells in that lineage
[119,120,121,122]	Mice lineage	Genetic differences in tuberculosis development between lineages	-Survival of infected animals -Bacterial burden in lungs, spleen and liver -Histological analysis of damaged lungs -Detection of quantitative trait loci (QTL) -Chemokine and cytokine production by lungs -Functionality of neutrophils	-Differences in survival data among different mice lineages -Differences in bacterial load in each lineage -Differences in pulmonary pathology displayed in each lineage -H-2^k^ haplotype is the more susceptible type of mice
[123,124,125,129,130,132,136,137,138,139,140,141]	Mice lineage	C3HeB/FeJ	-Survival of infected animals -Bacterial burden in lungs, spleen and liver -Histological analysis of damaged lungs -Polymorphisms mapping -Determination of cytokine mRNA profile -Determination of immune cells populations in the lungs	-C3HeB/FeJ develops granulomas with central necrosis [123] and hypoxia [125] after intravenous infection -Upon aerosol infection, this lineage develops different types of lesions [124] -Polymorphisms in *sst1* locus [129,131] and in chromosome 7 [132] are related with enhanced susceptibility -Neutrophils are responsible of generating the active tuberculosis-like lesions [136,137] -Host-directed therapies can be tested in this lineage [138,139] -Cording is a key virulence factor of *M. tuberculosis* [140]
[143,144,145,146,147,148,149]	Mice lineage	DBA/2	-Survival of infected animals -Bacterial burden in lungs, spleen and liver -Histological analysis of damaged lungs -Polymorphisms mapping -Determination of cytokine mRNA profile -Determination of immune cells populations in the lungs	-Dissemination of the infection is caused by migration of infected macrophages to uninfected pulmonary tissue [143] -*TRL3* and *TRL4* are related to susceptibility against the infection [145] -Macrophages from this lineage are less able to deal with the infection [146] -This lineage presents less dendritic and regulatory T cells than other lineages [147] -Mutants of *M. tuberculosis* in the synthesis of proline, tryptophan and mycolic acid methyltransferases showed reduced virulence [148,149]
[151,152,153,155,157,158,159,160,161,162,165,166]	Mice lineage	129/Sv	-Survival of infected animals -Bacterial burden in lungs, spleen and liver -Histological analysis of damaged lungs -Determination of cytokine mRNA profile -Determination of immune cells populations in the lungs -Macrophage isolation	-*Nramp1* [122,151] and Dectin-1 [152] have no effect on resistance to the infection -Nitric oxide [155,157], TNF-α, IFN-γ [158], IL-6 [159], TAP 1 [160] and TIR8 [161] are necessary to control the infection -*M. tuberculosis* can induce the overproduction of foamy macrophages [162] -Sirtuin 1 can avoid macrophages apoptosis and enhance the control of the infection [165,166]
[167,169,170,171,172,173,174,175,176,179,180,183,184,185]	Mice lineage	BALB/c	-Survival of infected animals -Bacterial burden in lungs, spleen and liver -Histological analysis of damaged lungs -Determination of cytokine mRNA profile -Determination of immune cells populations in the lungs	-Imbalance of the Th1/Th2 response leads to development of tuberculosis [167] -BALB/c offers a good latent tuberculosis model for drug testing [169,170,171,172,173] -IL-4 [174,175,176] and IL-12 [179,180] participate in the Th1/Th2 balance and in the control of the infection -*rv0180* [183] and *sigE* [184,185] are two new virulence mechanisms of *M. tuberculosis*
[186,187,188,189,190,191,192,193,194,195,196,197,200,201,202,205,206,207,208,209,214,215]	Mice lineage	C57BL/6	-Survival of infected animals -Bacterial burden in lungs, spleen and liver -Histological analysis of damaged lungs -Determination of cytokine mRNA profile -Determination of immune cells populations in the lungs -Macrophage isolation	-Nitric oxide plays an important role in controlling the infection [186,189] -C57BL/6 can also be used to recreate the latent tuberculosis scenario where therapeutic vaccines can be tested [180,181,182,183,184,185,186,187,188,189,190,191,192,214,215] -IL-10 [193], IL-12 [194], and IL-6 [195] are molecules needed to control the infection -microRNAs control several cellular processes related with susceptibility to the infection [200,201,202,205,206,207,208,209]
[216,217,218]	Unbalanced immunity	HIV and *M. tuberculosis* co-infection	-Bacterial and viral burden in lungs, spleen and liver -Histological analysis of damaged lungs -Determination of cytokine profile in the lungs -Determination of immune cells populations in the lungs	-Higher bacterial loads in mice co-infected with both agents -Disorganized granulomas and increased pulmonary pathology in animals co-infected with both agents -Increase in the number of neutrophils in the granulomas of co-infected animals -Increase synthesis of pro-inflammatory cytokines in co-infected animals
[219]	Unbalanced immunity	Influenza virus and *M. tuberculosis* co-infection	-Systematic review	-Co-infection with both agents worsened the course of tuberculosis
[220,223]	Unbalanced immunity	Coronavirus and *M. tuberculosis* co-infection	-Bacterial and viral burden in the lungs -Determination of immune cells populations and activity in the lungs	-Higher bacterial loads in mice co-infected with both agents -More permissive immune cells populations in co-infected animals
[222]	Unbalanced immunity	Coronavirus and *M. tuberculosis* co-infection	-Bacterial and viral burden in lungs, spleen and liver -Histological analysis of damaged lungs -Determination of cytokine profile in the lungs -Determination of immune cells populations in the lungs	-No effect of the co-infection upon bacterial load -No effect of the co-infection upon pulmonary damage -No effect of the co-infection upon cytokine profile -No effect of the co-infection upon immune cell populations
[227,228]	Unbalanced immunity	Parasitic and *M. tuberculosis* co-infection	-Bacterial and parasitic burden in lungs, spleen and liver -Histological analysis of damaged lungs -Determination of cytokine profile in the lungs -Determination of immune cells populations in the lungs	-Higher bacterial burden in animals co-infected with helminths -Higher pulmonary damaged area in animals co-infected with helminths -Biased immune response towards Th2 type
[229,230]	Unbalancedimmunity	Parasitic and *M. tuberculosis* co-infection	-Bacterial and parasitic burden in lungs, spleen and liver -Histological analysis of damaged lungs -Determination of cytokine profile in the lungs -Determination of immune cells populations in the lungs	-Non effect or reduced bacterial load in animals co-infected with helminths -Non effect or reduced pulmonary damaged area in animals co-infected with helminths -Biased immune response towards Th2 type
[233,234]	Unbalanced immunity	*Plasmodium* and *M. tuberculosis* co-infection	-Bacterial and parasitic burden in lungs, spleen and liver -Histological analysis of damaged lungs -Determination of cytokine profile in the lungs -Determination of immune cells populations in the lungs	-Reduced survival in co-infected animals -No differences in pulmonary damaged area -Higher bacterial load in co-infected animals -Biased immune response towards Th2 type or not efficient Th1 type
[237,238,239]	Unbalancedimmunity	Malnutrition (deficiency)	-Survival of infected animals -Bacterial burden in lungs, spleen and liver -Histological analysis of damaged lungs -Determination of cytokine mRNA profile -Determination of immune cells populations in the lungs, spleen and lymph nodes	-Reduced survival in malnourished mice -Higher bacterial load in malnourished mice -Poorly structured granulomas formed in malnourished mice -Initial diminution of nitric oxide, IFN-γ and TNF-α in malnourished mice -Increased CD4 and CD8 T cells numbers in malnourished mice
[242,243,244]	Unbalancedimmunity	Malnutrition (overnutrition)	-Survival of infected animals -Bacterial burden in lungs, spleen and liver -Histological analysis of damaged lungs -Determination of cytokine profile -Determination of immune cells populations in the lungs -Intestinal microbiota determination -Protection efficacy of BCG	-Reduced survival in obese mice -Higher bacterial load in obese mice -Higher pulmonary damage in obese mice -Increase in IFN-γ production and inflammatory responses in obese mice -Higher frequencies of proinflammatory CD4 T cells in obese mice -Changes in intestinal microbiota could be associated with increased susceptibility
[248,250,251]	Unbalancedimmunity	Diabetes	-Survival of infected animals -Bacterial burden in lungs, spleen and liver -Histological analysis of damaged lungs -Determination of cytokine profile -Determination of immune cells populations in the lungs -Macrophage isolation and functionality assays	-Reduced survival in diabetic mice -Higher bacterial load in diabetic mice -Higher pulmonary damage in diabetic mice -Discordances regarding cytokine profiles -No differences in the frequencies of T cell populations between groups -Reduced phagocytic capacity of macrophages from diabetic mice -Reduced T-cell promoting capacity of macrophages from diabetic mice

* potentially appliable to sex differences, as there is a sexual dimorphism in the steroid production.

## Data Availability

Non applicable.

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
