# Peer review of "Mouse Models for Mycobacterium tuberculosis Pathogenesis: Show and Do Not Tell"

_pathogens, 2022, doi:10.3390/pathogens12010049_

Round 1

Reviewer 1 Report

The authors have done a extensive job  of compiling the work carried out with murine models and TB.

It would be better to consider the following suggestions for betterment.

1.Too much historical content does not give any value to this manuscript.

2.There is no details about the various techniques, which can be perfromed with murine modes for various studies.

3.Applications of murine models for specific purpose like Therapeutics, immunology, histology, eptb, Transgenic models, knockout strains, kind of inclusions can be of much help.

4. scientific gaps and future knowledge where murine studies  for Tb can be helpful, that should be included.

Can cut down on lot of historic and descriptive details, for better citation and reach.

Author Response

-Too much historical content does not give any value to this manuscript.

The length of the introduction has been reduced in order make it more precise and focused.

-There is no details about the various techniques, which can be perfromed with murine modes for various studies.

Various corrections and updates, especially in the section 2.2. Role of mice in the study of tuberculosis development and in section 3. Conclusions have been added regarding this topic.

-Applications of murine models for specific purpose like Therapeutics, immunology, histology, eptb, Transgenic models, knockout strains, kind of inclusions can be of much help.

A paragraph regarding which types of lineages should be used in several tuberculosis scenarios, including diagnosis, therapeutics… has been added in section 3. Conclusions.

-scientific gaps and future knowledge where murine studies for Tb can be helpful, that should be included.

Some perspective and future studies have been added in the section 3. Conclusions.

Reviewer 2 Report

1. When we are talking about mice, it would be better to use the term lineage, but not strain

2. The description of some  aspects of mice use in veterinary medicine associated with TB study would be the benefit. 
3. The legislative area overview would be a benefit for this work. 
4. It would be great to add some perspective studies forecast of suggestions. 

Author Response

-When we are talking about mice, it would be better to use the term lineage, but not strain
Every “strain” (when talking about mice) has been substituted for “lineage”.

-The description of some aspects of mice use in veterinary medicine associated with TB study would be the benefit.
That was a very interesting topic to study and write about, unfortunately, the bast majority of the experimental assays related to that topic were focused in the study of Mycobacterium bovis pathogenicity and involved the use of bigger animal models, like goats or cows. As this review was focused mainly in the use of mice, we decided not to include this part in our review, although it would be useful in a possible review about animal models in general.

-The legislative area overview would be a benefit for this work.
A paragraph of legislative aspects in Europe and US has been added, including a small comparison regarding the definition of “laboratory animals” between one region and the other. This has been added as the second paragraph in section 2.2. Role of mice in the study of tuberculosis development.

-It would be great to add some perspective studies forecast of suggestions.
Some perspective and future studies have been added in the section 3. Conclusions

Reviewer 3 Report

Pablo Soldevilla et al. review the mouse models for mycobacterium tuberculosis pathogenesis, which will help biomedical researchers in the design of new tools for diagnosis, treatment and prevention of tuberculosis. This is an update to the mouse model study for mycobacterium tuberculosis. Overall, this is a common topic.

 1. “2.1. M. tuberculosis infectious pathway”, this section is important for the manuscript, so more of this section must be added.

2. “2.1. M. tuberculosis infectious pathway”, this section should be summarized using a new or separate Figure or table. I'd like to see more details.

3. This manuscript lacks focus. Age, sex, different breeds of mice are relatively not important. These sections should be appropriately reduced. Meanwhile, It is better to add more details to other sections.

4. Some perspective studies should be added.

5. English grammar should be further improved.

Author Response

-“2.1. M. tuberculosis infectious pathway”, this section is important for the manuscript, so more of this section must be added.
A more detailed paragraph about the destructive role of neutrophils in active tuberculosis scenarios has been added in this section. In addition, it has been added how M. tuberculosis is able to countermeasure the initial attack of surfactant once the bacillus arrives to the lung, the formation of cords inside alveolar macrophages and the development of Macrophage External Traps by macrophages as “recently” discovered new defense mechanism of the host.

-“2.1. M. tuberculosis infectious pathway”, this section should be summarized using a new or separate Figure or table. I’d like to see more details.
A new figure (Figure 1) has been created and added in response to this suggest.

-This manuscript lacks focus. Age, sex, different breeds of mice are relatively not important. These sections should be appropriately reduced. Meanwhile, It is better to add more details to other sections.
After a lot of consideration, and taking into account the comments of the other reviewers, we couldn’t find a way to reduce information regarding age, sex and genetic differences and please the other reviewers comments, so we decided not to reduce the information written regarding that aspects, although we understood the purpose of your suggestion. However, more information has been added to the section 2.2.4. Immune imbalance scenarios, in particular, a deeper implication of studies with animals and Coronavirus and the Tuberculosis-Malaria coinfection scenario. Table 1 has been updated with the new references. In addition, we have added more parts to the section 3. Conclusions that, in our opinion, make that sections more connected with the entire text.

-Some perspective studies should be added.
Some perspective and future studies have been added in the section 3. Conclusions.

-English grammar should be further improved.
The text has been reviewed by an internal member of the group and several corrections have been applied.